# Influence of Heat Waves on Daily Hospital Visits for Mental Illness in Jinan, China—A Case-Crossover Study

**DOI:** 10.3390/ijerph16010087

**Published:** 2018-12-30

**Authors:** Xuena Liu, Hui Liu, Hua Fan, Yizhi Liu, Guoyong Ding

**Affiliations:** 1Department of Health Statistics, School of Public Health, Taishan Medical University, Taian 271016, China; xuena_liu@163.com (X.L.); 15805380399@163.com (H.F.); yzliu@tsmc.edu.cn (Y.L.); 2Office of Asset and Laboratory Management, Shandong Yingcai University, Jinan 250104, China; huiliu_dd@163.com; 3Department of Epidemiology, School of Public Health, Taishan Medical University, Taian 271016, China

**Keywords:** heat waves, mental illness, case-crossover study

## Abstract

*Background:* Given that more frequent and intensive extreme heat events have been projected based on climate change modeling, it is of significance to have a better understanding of the association between heat waves and mental illnesses. This study aimed to explore the effects of heat waves on daily hospital visits for mental illness in the summer of 2010 in Jinan, China. *Methods:* A symmetric bidirectional case-crossover study was firstly conducted to determine the relationship between daily hospital visits for mental illness and heat waves in Jinan in 2010. Multifactor logistic regression analysis was then used to analyze the influencing factors for daily hospital visits for mental illness during the heat wave periods. *Results:* Multivariable analysis showed that the heat wave events were associated with an increased risk of mental illness. The largest odds ratios (ORs) of the heat waves for daily hospital visits for mental illness were 2.231 (95% confidence interval (CI): 1.436–3.466) at a 3-day lag, 2.836 (95% CI: 1.776–4.525) at a 2-day lag, 3.178 (95% CI: 1.995–5.064) at a 3-day lag, and 2.988 (95% CI: 2.158–4.140) at a 2-day lag for the first, second, third, and fourth heat waves, respectively. The elderly, urban residents, outdoor workers, and singles may be high-risk populations for developing heat wave-related mental illness. *Conclusions:* Our study has supported that there is a positive association between heat waves and hospital visits for mental illness in the study site. Age, home address, occupation, and marital status were associated with daily hospital visits for mental illness during the heat wave periods.

## 1. Introduction

Climate change is likely the biggest global health threat of the 21st century [1,2,3,4]. As reported by Intergovernmental Panel on Climate Change (IPCC), the surface temperature of the earth has significantly warmed in the past few decades and it has been accelerating at a rate far greater than projected [5]. Heat waves, as a global environmental problem, are usually characterized by extremely high temperatures for long duration, and humans are unable to adjust to such extreme conditions. At the same time, the frequency and intensity of heat waves have been continuously increasing as global temperature increases [1,6,7,8,9,10,11,12,13]. It is worth mentioning that increasing temperature has direct and indirect impacts on human well-being, including mental health. In 2007, the Fourth IPCC assessment clearly reported multiple adverse health effects caused by climate change, alluding to the fact that many critical effects are psychological [13]. It has been documented that heat waves may not only directly affect mental health under extreme weather events, but also disrupt social, economic, and environmental determinants that promote mental health at the community level [2,14]. In addition, there are a variety of reasons to believe that the people with a preexisting mental illness are particularly vulnerable to the ambient temperature. That is to say, extremely high temperatures could exacerbate psychiatric conditions [1,11,12,13]. For example, some psychotropic medications increase vulnerability to heat-related morbidity by affecting normal thermoregulation [2,12]. Adaptation strategies, for some behavioral and psychiatric reasons, may be less achievable for those with mental illness and thus the affected individuals may be unable to effectively protect themselves from the effects of heat [9,13].

Mental illness has already caused a significant burden to the public health system. A survey in Europe showed that 25.9% of the respondents said they had suffered from mental illness at some point, with 11.5% of suffering from mental illness within the last year [15]. Statistics in Australia also indicate an increase in reported mental illness, with 11% of persons reporting long-term mental or behavioral problems in 2004–2005, up from 5.9% in 1995 [12]. With the increasing competitive social atmosphere in China, the number of people suffering from mental diseases is also skyrocketing [16].

An epidemiological survey indicated that one fifth of Chinese adults suffer from mental illness [17]. Mental illness is likely to constitute the second greatest burden of non-fatal disease of the world by 2030, and a better understanding of the patterns of mental illness associated with heat waves is important for public health to identify the health consequences resulting from global climate changes [1]. However, the association of heat waves with mental illness has not yet been investigated extensively. Research on the association between heat waves and mental illness is insufficient to support decision making.

Relevant data show that Eastern China (including Jinan) is one of the areas hit by heat waves most frequently. High temperatures are disastrous in the summers of this region. Since the beginning of the 21st century, both the frequency and intensity of high-temperature weather in this region have noticeably increased [18]. The Shandong province was stricken by nine rare extreme weather events in 2010, among which heat wave events were also listed. In the summer of 2010, several deaths occurred in Jinan because of heatstroke. Even in late September the temperature in Jinan remained above 35 °C, noticeably postponing the coming of autumn in that year. Therefore, Jinan was selected as our study area. Jinan has a longitude of 117°00′ E and latitude of 36°40′ N, covering an area of 8177.21 km^2^ and hosting a population of 6,814,000 in 2010. Jinan has a temperate continental monsoon climate, with an annual average temperature 14.7 °C and annual average rainfall of 671.1 mm. Extreme heat in Jinan usually occurs between June and August. Mental illness is common in Jinan; therefore, patients already with mental illness might be more sensitive to heat waves and consequently have higher risks for hospital visits. It is worthwhile evaluating the effects of heat waves on hospital visits for mental illness. Thus, this study aims to provide a better understanding of the impacts of heat waves on mental health in the summer (1 June–31 August) of Jinan in 2010 and to provide suggestions for better adaptation to future extreme heat events. It will assist in identifying populations at risk and in providing evidence for decision makers in reducing risks for mental health clinics, public health, and social service sectors.

## 2. Materials and Methods

A case-crossover design was first conducted to investigate the association between heat waves and the daily hospital visits of mental illness in Jinan of 2010. In the case-crossover design, cases serve as their own control or referent at a different time period before or after the disease event [19]. The case-crossover design is useful when studying transient exposures and acute effects because fixed individual characteristics that do not vary with time are controlled by design [18]. Multifactor logistic regression analysis was then used to analyze the influence factors with daily hospital visits for mental illness during the heat wave periods.

### 2.1. Heat Wave Events

Up to now, no consistency has been found between the most sensitive heat indicators for different locations due to various characteristics of events in different geographic and climatic conditions [20]. China has a vast land with climates in different regions varying greatly. According to the China Meteorological Administration, each province and city could set the threshold temperature value of heat waves based on the local climate features. This study defined heat wave events according to the climate features of Jinan and the relevant information: daily maximum temperatures ≥35 °C lasting for no less than three days. Therefore, there were four episodes of heat waves observed during the study period (in a total of 14 days), from 14 to 17 June, 28 to 30 June, 4 to 7 July, and 29 to 31 July in 2010, respectively. By convention, in this study, the first day of the heat wave (the day when the heat wave first occurred) was recorded as Day 0, followed by Day 1, and so forth.

### 2.2. Case and Control Periods

The heat wave periods in the study area, i.e., the periods of 14 to 17 June, 28 to 30 June, 4 to 7 July, and 29 to 31 July in 2010, were selected as the exposure periods, respectively. The 1:3 symmetric bidirectional design was applied for selecting control days, matching the day of the week with a 3-week lead or lag time from when the case occurred (the control periods 24 May to 8 July, 7 June to 21 July, 13 June to 28 July, and 8 July to 21 August were used for control selection, respectively). This approach has been shown to control for time-invariant confounders, time trends, and seasonal variation in exposure as well as to produce unbiased results using conditional logistic regression models [21]. If one exposure day was included in a control period, then it was excluded from the control days. One exposure day was matched with four to six control days in this way. There were 20 days, 12 days, 18 days, and 18 days selected as control days for the first, second, third, and last heat wave events, respectively.

### 2.3. Data Collection

#### 2.3.1. Disease Surveillance Data

The daily disease data of mental illness in Jinan of 2010 were collected from the mental health center of Shandong province. The detailed information of the cases was recorded in the case registration report system of the center. The electronic records we extracted included: case number, gender, age, type of diagnosis, home address, occupation, marital status, contact information, and so on. The inclusion criteria of the cases were as follows:(1)The cases had to be of permanent residents of Jinan or individuals who had been living in Jinan for at least three months before the illness.(2)Mental illness included a range of conditions, both short-term or chronic, such as depression, anxiety, and schizophrenia [12]. These conditions differ in their etiologies, symptoms, effects, and treatment. However, these conditions are all characterized by alterations in thinking, mood, or behavior, and the associated distress or impaired functioning [1]. The mental illnesses included in our study were selected by the International Classification of Diseases codes for 10th Revision (ICD-10: F00–F99), as shown in Table 1.(3)The admission cases included in our study were the new hospitalized cases during the study period.(4)The new cases together with the recurrent cases during the study period were researched together.

#### 2.3.2. Meteorological Data

Daily meteorological data from 1 June to 31 August in 2010 in Jinan were obtained from the China Meteorological Data Sharing Service System [22]. The meteorological variables included daily maximum temperature (Tmax), daily average temperature (AT), daily minimum temperature (Tmin), daily average relative humidity (ARH), daily average wind velocity (AWV), and daily average air pressure (AAP).

### 2.4. Statistical Analysis

Firstly, a descriptive analysis was performed to report the characteristics of the cases and distribution of meteorological factors. Then the disease distribution for hospitalized cases according to ICD-10 during the study period was analyzed. The Wilcoxon two-sample test was adopted to compare the differences in the cases between the heat wave and non-heat wave period.

Secondly, ORs and 95%CIs were derived by using conditional logistic regression models with every day and several control days as one stratum. There were 4, 3, 4, and 3 stratums for the first, second, third, and last heat wave events that we analyzed, respectively. In every stratum, the daily hospital visits for mental illness in exposure day were selected as frequency. A dummy variable “y” was used to represent exposure and control periods using 0 and 1. In the multivariable analysis, we adjusted meteorological variables which may influence the daily hospital visits for mental illness. With the consideration of potential lagged effects on mental illness of the heat waves, lagged effects up to 5 days were calculated by conditional logistic regression analysis.

Thirdly, a multifactor logistic regression analysis was conducted to explore the influence factors with daily hospital visits for mental illness during the heat wave periods. Many factors were difficult to obtain for outpatient cases, so all the hospitalized cases during the exposure periods were served in the case group and all the hospital admissions in the remaining days were allocated to the control group. The factors we analyzed and their assignment cases are shown in Table 2. The significance level to eliminate variables was 0.15, and the significance level to select variables was 0.10.

## 3. Results

### 3.1. Descriptive Analysis for the Disease and Meteorological Data

Figure 1 shows the daily maximum temperature and daily hospital visits in the study period of 2010 in Jinan. A daily distribution of mental illness and meteorological factors during the heat wave periods and non-heat wave periods in Jinan is shown in Table 3. During the study period from 1 June to 31 August of 2010, a total of 19,569 notified mental illness cases were identified, of which 3573 cases were notified during the heat wave periods, accounting for 18.3% of the total reported cases in the study period. The number of male patients was slightly larger than female ones (11,423 vs. 8146). The highest risk group was the people over 65 years (the morbidity was 2.35%). As can be seen from Table 4, the most common type of mental illness during the study period was F20–F29 (schizophrenia, classification disorders, and delusional disorders), followed by F30–F39 (mood disorders). The Wilcoxon two-sample test showed that the number of daily cases of F00–F99, F20–F29, F30–F39, and F40–F49 were significantly different between the heat wave and non-heat wave periods (*p* < 0.05).

### 3.2. Correlation Analysis

Table 5 demonstrates the results of Spearman’s correlation analysis of meteorological factors. The results suggested that statistical correlation was existed between any two meteorological factors. Thus, all the meteorological variables including AT, ARH, AWV, and AAP were incorporated in the next analysis models to estimate the risk of heat waves on daily hospital visits for mental illness.

### 3.3. The Symmetric Bidirectional Case-Crossover Study

Figure 2 shows the estimated ORs of the four heat wave events on the effect of the daily hospital visits of mental illness from a 1-day to 5-day lag in the conditional logistic regression models. The results indicated that the heat waves significantly increased the daily hospital visits of mental illness in Jinan of 2010. The strongest effect was shown with a 3-day lag for the first heat wave event (OR = 2.231, 95% CI: 1.436–3.466), a 2-day lag for the second heat wave event (OR = 2.836, 95% CI: 1.776–4.525), a 3-day lag for the third heat wave event (OR = 3.178, 95% CI: 1.995–5.064), and a 2-day lag for the last heat wave event (OR = 2.988, 95% CI: 2.158–4.140), respectively.

### 3.4. The Multifactor Logistic Regression Analysis

The results of the previous analysis showed that the optimal lag periods for the effects of the four heat waves on mental illness were 3 days, 2 days, 3 days, and 2 days, respectively. Accordingly, the dangerous periods for the four heat waves were 7 days, 5 days, 7 days, and 5 days, respectively. The remaining days (68 days) during the study period were severed as the control period (Table 6).

The result of the multifactor logistic regression analysis is showed in Table 7. It indicated that old people had a larger risk to develop heat wave-related mental illness than young people (OR = 3.034, 95% CI: 1.802–5.139). The risk of heat waves on mental illness in urban areas was more serious than that in the rural or suburban areas (OR = 1.523, 95% CI: 1.120–2.074). Outdoor workers (OR = 1.714, 95% CI: 1.198–2.398) and singles (OR = 1.709, 95% CI: 1.233–2.349) were more likely to suffer from mental illness during the heat wave periods.

## 4. Discussion

This study has quantified the association between heat waves and daily hospital visits of mental illness in 2010 in Jinan City, China. Our results indicate that heat waves play an important role in the hospital visits of mental illness during the summer. The study supports that the occurrence of heat wave events is associated with increased mental health problems, which has been reported in some countries. For example, one study in Australia pointed out that the number of people with mental illness is estimated to increase by 6–11% during a heat wave [1]. A study carried out in five regions of New South Wales demonstrated that the emergency hospital admission due to mental health with extreme heat events increased significantly during the prolonged heat events (OR = 1.07; 95% CI: 1.00–1.15) [20]; another Australian study reported that the total mental disease (ICD-9, 290–294-9, 580–5999; ICD-10, N00–N39) admissions increased by 7% (95% CI: 1–13%) during heat waves [23]. A study in France found that various diseases were associated with extreme heat, and for people aged below 60, 41% of mental disease was essentially related to extreme heat, while for those aged above 60, 30% of mental disease is associated with extreme heat [24]. Our study found that the number of cases of F00–F99, F20–F29, F30–F39, and F40–F49 during the heat wave periods is significantly higher than that during non-heat wave periods. Analysis from this shows that extreme heat not only has an impact on the daily hospital visits of mental illness, but also has an impact on the types of mental illness.

This study has identified a lagged effect of heat waves on hospital visits of mental illness. Studies have found that heat waves do not only affect the daily hospital visits of mental illness for the day, but also influences the daily hospital visits of mental illness for the next few days, i.e., a “lagged effect” is observed and the delayed effect usually takes place in the following 1–5 days [25,26,27]. This is exactly the reason why this study chooses to analyze the lagged effect of 0–5 days. That is to say, heat waves were significantly associated with an increased risk of mental illness in the 1–3 days delay, of which the optimal lag period was 3 days, 2 days, 3 days, and 2 days for the first, second, third and fourth heat wave event, respectively. The results imply that appropriate precautions should be taken not only on the day of the heat wave, but also on the next two to three days. Indeed, this is crucial to minimize the effect of heat waves on mental health.

As suggested by several studies, the heat-related effects are more remarkable in urban areas than that in the suburbs and the countryside [26,27,28,29,30]. For instance, extensive analysis of the Chicago heat wave of 1995 revealed that most of the victims were urban dwellers. As argued by Conti et al., the effects of heat waves on health can be serious for elderly persons, especially those in urban areas [25]. Similar findings have been reported in our study. One possible reason is that cities tend to be warmer than the surrounding suburbs. There are multiple reasons for this, but one of the significant reasons is the “urban heat island” effect (UHI). This effect occurs because the built structures such as concrete, asphalt, and metal preferentially absorb heat and it is then reradiated, thereby causing urban areas to be 5–11 °C warmer than surrounding rural regions [10]. Another study mentioned that, relative to the surrounding rural areas, the UHI effect can add 12–16 °C to ambient air temperature, which has been epidemiologically linked to excess mortality [27]. Another possible reason is that air pollution which may be associated with the occurrence and effects of heat waves is more severe in cities [31]. One study found that excess mortality during the 1995 heat wave in Greater London was markedly higher than for England and Wales as a whole, and it may be because of higher temperatures (particularly the higher temperatures associated with the UHI effect) as well as higher levels of air pollutants [32]. As suggested by the study result, the risk of heat waves on mental illness in urban areas was more serious than the rural or suburban areas (OR = 1.523, 95% CI: 1.120–2.074), which was consistent with previous studies.

Many studies have documented that the increasing risk of heat waves poses a threat to the elderly [3,20,24,33,34], particularly those in the 65-year-old [24,35] and 75-year-old age groups [20,34]. This is likely a result of the imbalance and damage in their body’s thermoregulation, as well as their weakened ability to adopt protective behavior. This study indicated that old people (≥65) had a high risk to develop heat wave-related mental illness than young people (OR = 3.034, 95% CI: 1.802–5.139). Hence, the management and protection of the elderly should be strengthened during the heat wave periods.

So far, few studies have identified whether occupational differences remain as a factor impacting the effect during the heat wave. A study pointed out that the groups exposed to burning sunlight are more easily affected by temperature and heats wave than other people [36]. In the hot summer, especially for workers of enterprises where workplaces are in the open air, individuals are confronted with serious exposure and high temperatures in the daytime. On the one hand, heatstroke is more likely to occur as they are physically suffering from the high temperature; on the other hand, the physical discomfort can easily affect their emotions and feelings, causing anxiety, enervation, low efficiency, etc., which are symptoms described as forming part of “emotional heatstroke” by psychologists [37]. The reasons above might increase the incident rate of mental unhealthiness among the open-air workers during heat waves. The multi-factor logistic regression analysis in this study reveals that outdoor workers, compared to other occupations, are more easily influenced by high temperatures and heat waves during these periods (OR = 1.714, 95% CI: 1.198–2.398); this highlights the importance of strengthening the precaution and protection for outdoor workers. For example, except for preparing food and medicine to cope with the heat and temperature, it also important to prepare them mentally with a sense of cooling to improve their capability for withstanding high temperatures A rational and scientific working system is important for outdoor workers so that they do not have to work during the high-temperature periods to maximally avoid or reduce the threat and harm of heat waves upon this group of people.

The role of marital status in the heat wave’s influence has not been widely studied yet. Canoui-Poitrine et al. found that single men are faced with a higher death rate during heat waves [38]. The result of the study found that single people (OR = 1.709, 95% CI: 1.233–2.349) were more likely to suffer from mental illness during the heat wave periods. The possible explanation is that the single people has already carried a huge psychological burden, bringing the potential risks of mental illness. High temperatures and heat waves could work as “catalyst” or “accelerant”, contributing to or accelerating the rise of psychological problems. Therefore, we need to pay more attention to the mental healthiness of the single, offering more care, understanding, and support, especially during heat wave days.

Three limitations of this study must be acknowledged.
(1)One of the limitations of the study was to include all diagnoses (ICD 10: F00–F99) as a whole. As the health problems belonging to F00–F99 have totally different biological and causal backgrounds, more studies are still needed to assess the risk of heat waves on morbidity of mental illness in future.(2)This study selected the daily hospital visits for mental illness as an acute effect index to measure the temperature’s influence on the mental illness. The study objects were mostly collected from outpatient cases. Compared with hospitalized cases, outpatient cases are hard to indicate exhaustive demographic information of the study objects, not allowing a further stratified analysis of the objects based on different demographic features.(3)The study only explored the influence of heat waves on hospital visits of mental illness in Jinan. Since the intensity and features of heat waves may vary between different regions and people from different areas may develop varied adaptive capacities to the extreme high-temperature weather, the conclusions drawn in this study are still limited geographically.

## 5. Conclusions

In conclusion, our study has confirmed that heat waves are positively associated with the risk of hospital visits for mental illness in the selected study area, with various lagged effects. The elderly, urban residents, outdoor workers, and singles may be more vulnerable to developing heat wave-related mental illness. Further research is necessary to assist evidence-based decision-making with regards to reducing the mental health burden associated with extreme heat.

## Figures and Tables

**Figure 1 ijerph-16-00087-f001:**
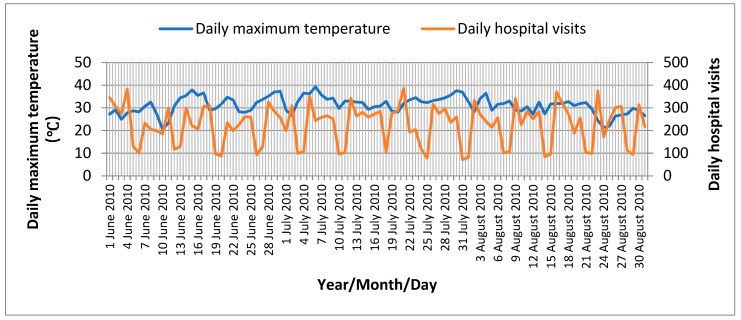
Daily maximum temperature and daily hospital visits in the study period of 2010 in Jinan.

**Figure 2 ijerph-16-00087-f002:**
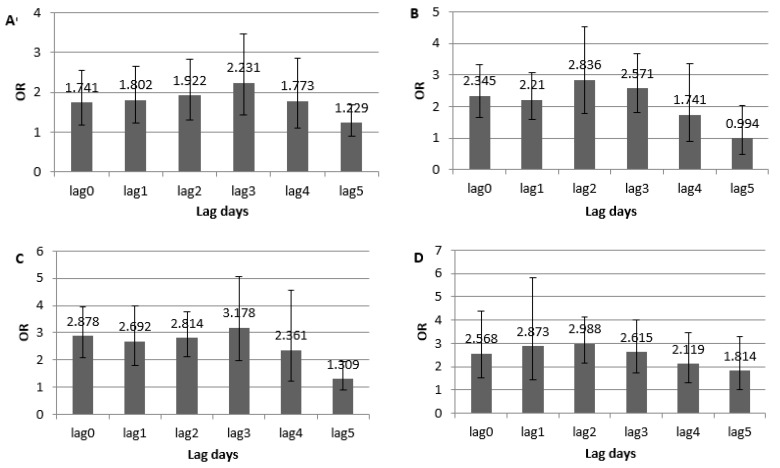
Odds ratio (OR) estimates of the heat waves on the hospital visits of mental illness in different lag days in Jinan. (**A**) the first heat wave; (**B**) the second heat wave; (**C**) the third heat wave; (**D**) the fourth heat wave.

**Table 1 ijerph-16-00087-t001:** Disease types of the International Classification of Diseases (ICD 10: F00–F99).

Codes	Disease Type
F00–F99	Mental and behavioral disorders
F00–F09	Organic (including symptomatic) mental disorders
F10–F19	Mental and behavioral disorders caused by the use of psychoactive substances
F20–F29	Schizophrenia, classification disorders, and delusional disorders
F30–F39	Mood disorders
F40–F49	Neurological, stress-related and physical disorders
F50–F59	Complex behavioral disorders associated with physiological disorders and somatic factors
F60–F69	Adult personality and behavioral disorders
F70–F79	Developmental disorders
F80–F89	Mental development disorders
F90–F98	Behavior and mood disorders are usually associated with childhood and youth
F99	Mental disorders

**Table 2 ijerph-16-00087-t002:** The factors we analyzed and their assignment cases.

Factors	Variable Name	Assignment Cases
Gender	X1	male = 1, female = 0
Age	X2	≥65 = 1, ≤64 = 0
Home address	X3	Urban = 1, rural or suburban = 0
Occupation	X4	outdoor workers = 1, indoor workers = 0
Marital status	X5	others(singles) = 1, married = 0

**Table 3 ijerph-16-00087-t003:** Description of daily mental illness cases and meteorological factors during the study period.

Variables	$\bar{x}\pm s$	Min	P25	P50	P75	Max
Heat wave period						
AT (°C)	31.3 ± 1.3	28.6	30.3	31.9	32.2	33.1
AAP (hpa)	981.2 ± 1.4	979.3	980.5	981.5	983.0	983.9
ARH (%)	50.1 ± 10.2	31.0	42.8	49.5	59.5	65.0
AWV(m/s)	3.0 ± 0.8	1.9	2.4	2.9	3.6	5.0
Daily hospital visits	248 ± 51	162	210	226	292	360
Non-heat wave period						
AT (°C)	25.8 ± 3.1	17.7	23.7	26.2	27.9	32.3
AAP (hpa)	988.0 ± 4.1	979.2	985.1	987.5	991.1	996.6
ARH (%)	70.6 ± 14.7	30.0	60.0	73.0	82.5	96.0
AWV (m/s)	2.1 ± 0.7	0.8	1.6	2.0	2.6	3.9
Daily hospital visits	215 ± 68	85	179	234	261	301

$\bar{x}\pm s$: mean ± standard deviation; Min: minimum; P25: the 25th percentile; P50: the 50th percentile; P75: the 75th percentile; Max: maximum; AT: daily average temperature; ARH: daily average relative humidity; AWV: daily average wind velocity; AAP: daily average air pressure.

**Table 4 ijerph-16-00087-t004:** Description of daily hospitalized cases during the study period according to ICD-10.

Codes	Heat Wave Period		Non-Heat Wave Period		*p*-Value
	*n*	$\bar{x}\pm s$	*n*	$\bar{x}\pm s$	
F00–F99 *	238	17.03 ± 0.61	701	8.84 ± 0.53	0.016
F00–F09	7	0.50 ± 0.52	21	0.27 ± 0.59	0.072
F10–F19	19	1.29 ± 0.81	61	0.88 ± 1.03	0.349
F20–F29 *	108	8.06 ± 0.64	394	4.72 ± 0.90	0.028
F30–F39 *	57	4.12 ± 0.58	122	1.50 ± 0.67	0.041
F40–F49 *	47	3.49 ± 1.14	69	0.74 ± 0.86	0.035
F50–F59	0	0.00 ± 0.00	16	0.21 ± 0.34	-
F60–F69	0	0.00 ± 0.00	18	0.23 ± 0.53	-
F70–F79	0	0.00 ± 0.00	0	0.00 ± 0.00	-
F80–F89	0	0.00 ± 0.00	0	0.00 ± 0.00	-
F99	0	0.00 ± 0.00	0	0.00 ± 0.00	-

n: the number of cases during the study period; $\bar{x}\pm s$: mean ± standard deviation; * *p* < 0.05.

**Table 5 ijerph-16-00087-t005:** Results of correlation analysis of the meteorological factors.

Meteorological Factors	AT (°C)	ARH (%)	AWV (m/s)	AAP (hpa)
ARH (%)	−0.553 *	1.000		
AWV (m/s)	0.298 *	−0.400 *	1.000	
AAP (hpa)	−0.675 *	0.237 *	−0.280 *	1.000

AT: daily average temperature; ARH: daily average relative humidity; AWV: daily average wind velocity; AAP: daily average air pressure; * *p* < 0.05.

**Table 6 ijerph-16-00087-t006:** The exposure periods of heat wave and their corresponding dangerous periods and duration.

Heat Wave Events	Exposure Period	Duration of Exposure Period (d)	Lag Days (d)	Dangerous Period	Duration of Dangerous Period (d)
First	14 June to 17 June	4	3	14 June to 20 June	7
Second	28 June to 30 June	3	2	28 June to 2 July	5
Third	4 July to 7 July	4	3	4 July to 10 July	7
Fourth	29 July to 31 July	3	2	29 July to 2 August	5

**Table 7 ijerph-16-00087-t007:** Results of the influence factors of mental illness during heat waves of the multivariate logistic regression.

Influence Factors	B	S.E.	Wald	*p-*Value	OR	95% CI
Gender	0.051	0.040	1.783	0.201	1.057	0.972,1.145
Age	1.203	0.271	16.966	0.000	3.034	1.802,5.139
Home address	0.420	0.157	7.199	0.007	1.523	1.120,2.074
Occupation	0.529	0.168	9.016	0.003	1.714	1.198,2.398
Marital status	0.536	0.172	10.366	0.001	1.709	1.233,2.349

B: regression coefficient; S.E.: standard error; Wald: Wald χ^2^; Gender: male or female, reference: female; Age: ≥65 or ≤64, reference: ≤64; Family address: urban or (rural or suburban), reference: rural or suburban; Occupation: outdoor workers or indoor workers, reference: indoor workers; Marital status: married or others, reference: married. CI: confidence interval.

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
