# Peer review of "Influence of Heat Waves on Daily Hospital Visits for Mental Illness in Jinan, China—A Case-Crossover Study"

_ijerph, 2018, doi:10.3390/ijerph16010087_

Round 1

Reviewer 1 Report

Influence of heat waves to daily hospital visits for mental illness in Jinan, China - a case-crossover study

Generally, it is an interesting study, however, the methods used and the presentation of the result could be greatly improved.

Abstract: When reporting the results, instead of writing three repetitive sentences include all three ORs & co in the same sentence: “… 2.231 (95% CI: 1.436–3.466), 3.178 (95% CI: 1.995–5.064) and 2.988 (95% CI: 2.158–4.140) for the first, second and third heatwave respectively…” When it comes to conclusions, it sounds too strong to use the expression of “our study has confirmed” – say instead that you “found evidence for” or that you study “supported that…”

Introduction: You take up important background information, however you could structure it better (begin with overall health effects and continue then to the specific mental health effects; do not jump forth and back between these). Furthermore, your way to use the English language is occasionally rather naïve and needs to become a bit more scientific.

Methods and Results: The whole Methods part needs to be restructured – Right now, you include a lot of information that should be given in the background (Introduction) or in the discussion. Begin with the description of the area and population, continue to data collection and statistical analyses. Do not repeat the same information in several places!!!  Also, where (geographically and in relation to the study population) were the meteorological data measured?

The rationale of the dummy variables is somewhat surprising… You only divide the population to farmers or non-farmers… which is very unspecific… if you can only make broad assessments it would be better to use outdoor- versus indoor workers instead. Also, are there only two marital stages (married and divorced)??? Where are all the single, widowed and so on people? Why not generalize to married versus single instead?

One of the more important weaknesses is though, if I understood it correctly, that you have included all diagnoses (ICD 10: F00-F99) into the same model! As the health-problems belonging to F00-F99 have totally different biological and causal backgrounds, it would be so much better to separate these into groups, and maybe also exclude some of these.

Discussion: Again, the content of the text is relevant but repetitive and the language slightly naïve.

Reviewer 2 Report

My relatively minor comments and suggestions to further strengthen the manuscript are as follows:

Title

·         Would suggest rephrasing title to read “Influence of heat waves on daily hospital visits…”

Abstract

·         (Line 13): it is not clear what “insensitive” means with regard to extreme heat events; suggest rewording

Introduction

·         (last paragraph): would benefit from including background mental health statistics of relevance to the study country (China) or region (Jinan City)

Materials and Methods

·         (Line 128): “(3) The admission cases included in our study were the news during the study period.” - “the news” would benefit from rewording

·         (Line 130): delete subheading “2.3. Statistical analysis”

Discussion

·         (Line 168): could you comment on the unusually high proportion of people aged ≥65 years (74.89%) among mental health admissions following heatwaves (and to what extent this may be reflective of the broader age composition of the population in the Jinan area)

·         To inform future preparedness planning, it would be useful to know which specific mental disorders people were admitted for following heatwaves (and also, if the profile of mental disorders following heatwaves differed in any notable way from the mental disorder profile of routine admissions)

·         (Line 273): “described as emotional heatstroke by psychologists” would benefit from a reference

·         (Line 80): in the Introduction you indicate that five other extreme events may have occurred in Jinan during the summer of 2010 (in addition to the four heatwaves that form the focus of this study) – could you briefly mention what these events were, and consider their possible impact (if any) on hospital admissions for mental illness during the study period

Other comments

·         The manuscript would benefit from minor language editing

·         The reference list requires some formatting to adhere to the journal style (e.g., consistent use of abbreviated journal names)

Round 2

Reviewer 1 Report

Thank you for the responses and comments, however, I am still rather concerned about your attempt to clump all cases of mental illness, regardless of their classification (for example developmental effects, dementia, substance abuse and depression) together as one! What is the rationale behind this? Did you discuss prior the analyses the possible biological effects that heat is likely to have on different diagnoses? At least there should be some information about how many cases of each type of illness happened during the study period.
